# Occurrence of Trace Heavy Metals in Leaves of Urban Greening Plants in Fuxin, Northeast China: Spatial Distribution & Plant Purification Assessment

Qili Yang [1,*], Jing Guo [1], Dongli Wang [1], Yong Yu [2], Weili Dou [1], Zhiwen Liu [1], Qiaohong Xu [1] and Gang Lv [1]

[1] School of Environmental Science and Engineering, Liaoning Technical University, 47 Zhonghua Road, Fuxin 123000, China; guojing8891@163.com (J.G.); starhome0522@163.com (D.W.); dwl230024@163.com (W.D.); liuzhiwen0531@163.com (Z.L.); xuqiaohong0806@163.com (Q.X.); lvgang2637@126.com (G.L.)
[2] Key Laboratory of Wetland Ecology and Environment, Northeast Institute of Geography and Agroecology, Chinese Academy of Sciences, Changchun 130022, China; yuyong@iga.ac.cn
* Correspondence: yangqili@lntu.edu.cn

**Abstract:** Trace element analysis, in the leaves of five kinds of greening plants (*Buxus*, *Picea*, *Pine*, *Juniperus* and *Platycladus*) from eight uniform distribution sites in Fuxin, a typical traditional resource-based city in northeast China, was carried out to study the purification ability difference of urban greening plants and spatial distribution tendency of heavy metal elements in the whole city area. In terms of the purification ability analysis, *Platycladus* had a better environmental purification capacity for Cd, As, Pb and Cr. *Juniperus* also showed a certain environmental purification potential for As, Pb and Cu. Furthermore, Mn has the highest point mean of element content in all plants, ranging from 64.044–114.290 μg/g, and the MnPA content of *Buxus* and *Juniperus* was 60% higher than that of the other three plants, which showed a better Mn purification effect. In terms of the spatial distribution tendency analysis, point pollution source location and the urban climate factors (mainly for the wind factor) were the main controlling factors. However, the specificity of Mn distribution suggested that its polluting behavior had a close relation with minerals transportation during exploiting and transferring in the city's coal mining industry in the past.

**Keywords:** environmental indicator; ICP-MS; source analysis; urban greening plant

## 1. Introduction

Urban greening plants play an important role in urban environmental purification [1–4]. They purify the surrounding air through their own growth and metabolism process, and store pollutants in their biological bodies [5–8]. Leaf is the direct tissue where green plants carry out photosynthesis and respiration, and the main part of material exchange between the plants and surrounding [9–11]. During this process, some pollutants in the ambient air also enter the leaf tissue [12–14]. Therefore, plant leaves can be used as environmental indicator, which record the occurrence information of some environmental pollutants around the plant habitat when plants grow.

Air pollution in the urban environment is widely recognized as one of the most harmful threats to human health [15–17]. Heavy metals are an important concern in urban air quality monitoring [18–23] because they usually can make a series of threat to human health easily [24–26]. However, it is difficult to accurately determine the environmental impact information due to their low contents in air samples, high cost of the monitoring instruments and extensive sampling in space and time [26,27]. Plant leaves continuously exchange substances with air during the growth process [28–30], air pollutants can be taken up by plant leaves via stomata [26,31]. In the long-term accumulation process, low contents of heavy metals in the air can be accumulated in plant tissues, so as to provide a good sample record for the studies of air pollution around the plant habitat [26,27,32]. In the Metropolitan Area

of Rio de Janeiro City (Brazil), *Nerium oleander* L. leaves were conformed to be an efficient biological indicator for environmental pollution analysis since it is precise, fast and low-cost, besides allowing the monitoring of pollution levels over time [26]. Nadgórska-Socha et al. [33] studied the levels of heavy metal contamination on plants growing in urban biotopes in the city of Dąbrowa Górnicza (southern Poland) and found the mean values of APTI (Air Pollution Tolerance Index) for the investigated plants were in the following ascending order, *P. lanceolata < R. pseudoacacia < B. pendula < T. officinale*. Dafré-Martinelli et al. [9] reported a research result about the leaf accumulation level of trace elements of three most abundant native tree species (*Astronium graveolens, Croton floribundus* and *Piptadenia gonoacantha*) from remnants of Semideciduous Atlantic Forest, which was surrounded by numerous industries, intense road traffic and agricultural lands in Southeast Brazil. They concluded that *A. graveolens* was the most appropriate species to discriminate spatial variations and the forest closer to the industrial area was distinguished from the others by Mn and Ni from oil burning. This research revealed plant leaves could be applied for monitoring atmospheric pollution.

Fuxin is located in northwest Liaoning Province (Figure 1) and has been known for its coal mining industry until recently. Over recent decades, it enjoyed a reputation as a 'Coal and Power City', possessing Asia's largest open-cast mine and most productive thermal power station [34,35]. With the gradual exhaustion of coal resources and the transformation of urban economic structure, Fuxin is experiencing various development difficulties experienced by many traditional resource-based cities [36–38], and environmental quality must be an important consideration. It is of great significance to study the urban atmospheric environmental quality in the post-mining era, especially the background and distribution characteristics of heavy metal air pollution related to mineral resources for controlling and improving the urban environmental quality.

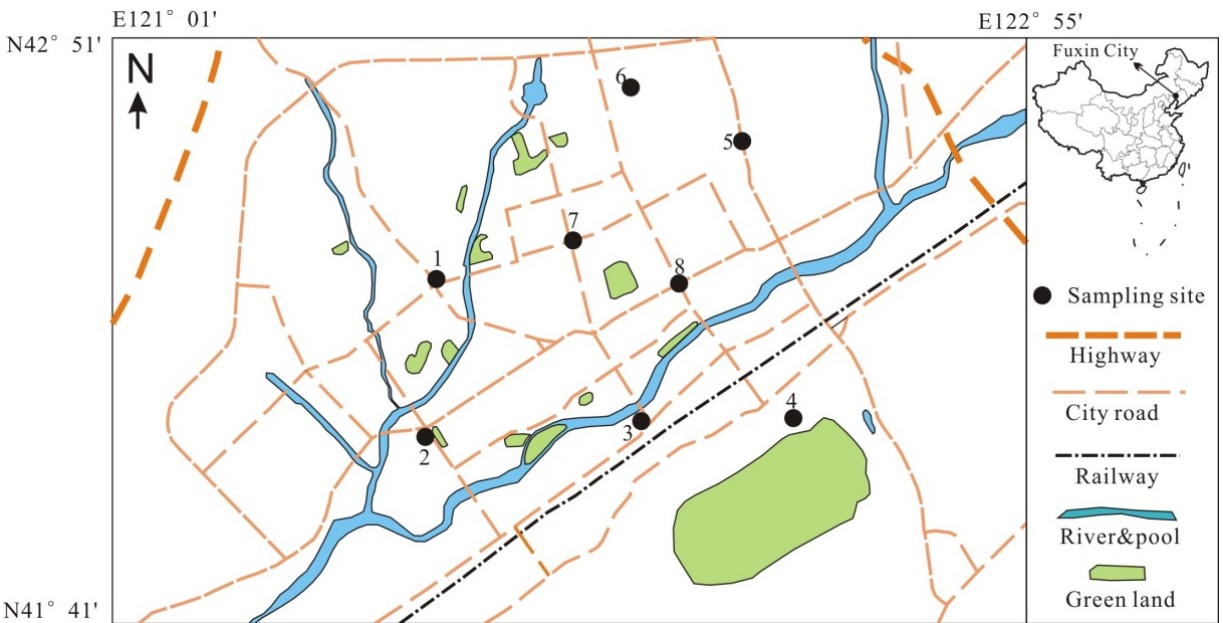

**Figure 1.** Sampling area and location in Fuxin.

## 2. Materials and Methods

### 2.1. Sampling and Reagents

Fuxin is located in northwest Liaoning Province of China, it is bounded between longitude 121°01′ and 122°55′ east and latitude 41°41′ to 42°51′ north. As shown in Figure 1, 8 fairly uniform points with good growth of green plants near urban traffic roads were set as sampling sites, and leaf tissues of 5 common green plants (*Buxus, Picea, Pine, Juniperus, Platycladus*) were collected at each site. The collected leaves were labeled and stored in paper bags, and the samples were brought back to the laboratory for air drying. A total of 40 samples were collected, with each sample about 5 g. All leaf samples were

washed with deionized water to clean the dust on the surface, and then dried and ground with an 80-mesh sieve.

All reagents used in this study were high grade (chromatographically pure). HCl (36–38%), $HNO_3$ (65–68%), HF (40%) and $H_2O_2$ (33%) were acquired from Beijing Chemical Inc., Beijing, China. All trace element reference materials were purchased from China National Nonferrous Metal and Electronic Materials Analysis and Testing Center, Beijing, China.

### 2.2. Sample Analysis

Aliquots of 200 mg of plant powder were wetted with 1 mL deionized water, then added 8 mL of nitric acid, 2 mL perchloric acid, 1 mL hydrofluoric acid for soaking overnight. After that all samples were digested at 120 °C for 2.5 h and 210 °C for 3 h successively. Then digestion liquid was diluted to 50 mL with 2% dilute nitric acid. After the solution was filtered by water filtration membrane, 1 mL was taken, and the volume was diluted to 10 mL with 2% nitric acid for ICP-MS determination. The concentrations of 8 trace elements (As, Mn, Cd, Ni, Cr, Pb, Cu and Zn) were determined by an ICP-MS (NexION 350X, PerkinElmer, Inc., Shelton, CT, USA).

### 2.3. Statistic Analysis (SA)

Statistical analysis was performed using IBM SPSS Statistics 19.0 (IBM Inc., Almont, NY, USA)and Microsoft Office 2007 Excel (Microsoft Inc., Redmond, WA, USA). Oneway ANOVA analysis was used to investigate the relationship between sampling sets and contents of each metal. Statistical significance was determined at the $p = 0.05$ level. So, we performed two sets of ANOVA one for the parameter "metal" between sites and one for the parameter "metal" between plants.

### 2.4. Quality Control (QC)

Reagent blanks and analytical duplicates were used for ensuring the accuracy and precision of analysis. Glass wares were soaked overnight with $HNO_3$ (10% $v/v$) and rinsed thoroughly with deionized water. The standard reference material (GBW 07405 [GSS-5]) obtained from the Center of National Standard Reference Material of China was used in the digestion and determination as part of QA protocol. The detected values of As, Mn, Cd, Ni, Cr, Pb, Cu and Zn in reference materials were within their certified concentration ranges. The recovery rates for the seven observed metals were around 90–110%. The limits of determination (LOD) of As, Mn, Cd, Ni, Cr, Pb, Cu and Zn were 0.6, 0.07, 0.9, 0.4, 0.2, 0.04, 0.2 and 0.3 ng/L, respectively.

### 2.5. $M_{SA}$ and $M_{PA}$

To compare the occurrence of trace elements in vegetation at different sampling sites, we used the mean element content of each plant at a sampling point, namely metal concentration of species average ($M_{SA}$), to represent the occurrence level of trace element M by plants at that point:

$$M_{SA} = \Sigma C_{Mij}/i \quad (M = \text{metal style, i = species number, j = sampling point number}) \quad (1)$$

Meanwhile, to compare the levels of trace elements in different plants, we used the average value of elements in each sampling site of a certain plant, namely metal concentration of point average ($M_{PA}$), to indicate the level of trace element M in this plant:

$$M_{PA} = \Sigma C_{Mij}/j \quad (M = \text{metal style, i = species number, j = sampling point number}) \quad (2)$$

## 3. Results and Discussion

### 3.1. Trace Elements from Greening Plants Leaves

As shown in Figure 2, the concentrations of eight trace elements in the leaves of different green plants showed notably different ranges. The content of As is 0–11.762 µg/g, Cd is 0.111–0.623 µg/g, Cr is 0–117.828 µg/g, Cu is 7.465–145.452 µg/g, Mn is 26.062–326.248 µg/g.

The contents of Ni are ranged from 0.364–49.921 µg/g, Pb is 0.822–13.261 µg/g and Zn is 27.549–131.593 µg/g. The relative content differences of trace elements at different points in different vegetation types are shown in Figure 2. The occurrence of different trace elements in the sample is affected by many factors and many processes. Here it is necessary to consider the most direct factors (spatial location and species of greenery) in different dimensions. One-way ANOVAs analysis was applied to assess these content differences of leaf trace elements in both of them. Their results showed in following Table 1.

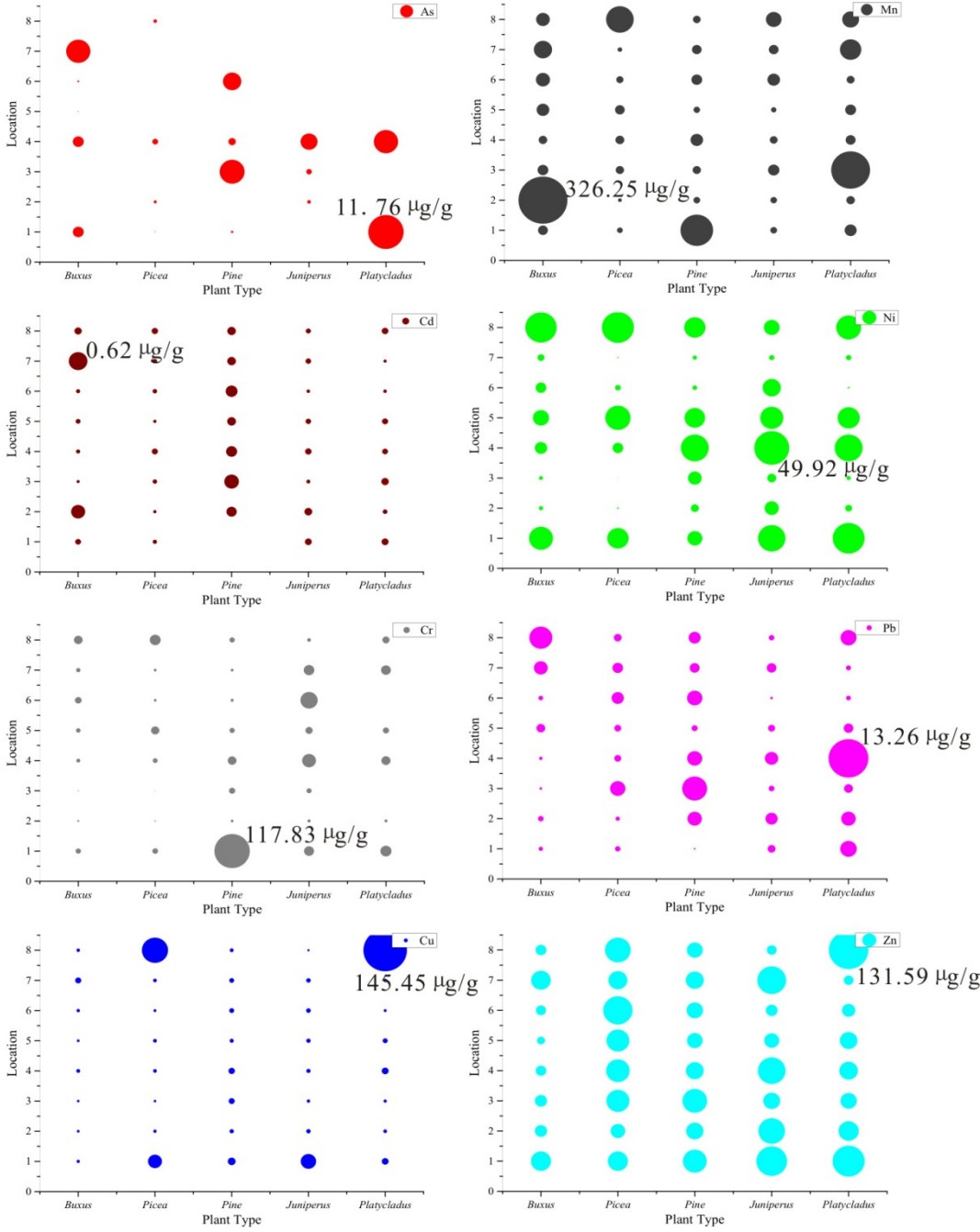

**Figure 2.** Trace element distribution from leaves of five kinds of greening plant in eight sites (Five kinds plant: *Buxus*, *Picea*, *Pine*, *Juniperus* and *Platycladus*; eight sites: as shown in Figure 1; bubbles' scale represented the concentration of metal and the max value were marked nearby).

**Table 1.** One-way ANOVAs results about content differences of leaf trace elements.

| Factor | Parameter | As | Cd | Cr | Cu | Mn | Ni | Pb | Zn |
|---|---|---|---|---|---|---|---|---|---|
| Spatial | F | 1.953 | 0.331 | 1.965 | 2.008 | 0.506 | 8.922 | 0.644 | 0.831 |
| Location | p | 0.093 | 0.934 | 0.091 | 0.085 | 0.823 | **0.000** | 0.717 | 0.570 |
| Greenery | F | 0.856 | 4.931 | 0.978 | 0.745 | 1.563 | 0.516 | 1.859 | 2.451 |
| Species | p | 0.500 | **0.003** | 0.432 | 0.568 | 0.206 | 0.725 | 0.140 | 0.064 |

Statistical results showed that there are significant differences for leaf Ni content in spatial position ($p < 0.05$) and for leaf Cd content in various plant species ($p < 0.05$). For more detail acquirement about spatial and specie distribution pattern of leaf trace element, further analysis should be proceeded in both directions (spatial and species) separately.

*3.2. Spatial Variation of Trace Element Contents*

The occurrence of trace elements in vegetation at different sampling sites were estimated using $M_{SA}$ value.

As shown in Table 2, the species average content of all 8 trace elements varied a lot at different sampling sites. For $As_{SA}$, it was significantly higher at site 1 and 4, lower at site 2, 5 and 6. There was little difference in $Cd_{SA}$ content at each point. $Cr_{SA}$ content was significantly higher at site 1, significantly lower at site 2 and site 3, and similar at other sites. $Cu_{SA}$ content was significantly higher at site 8, followed by site 1, and lower at other sites. $Mn_{SA}$ content was higher at site 2, 3 and 8, but lower at other sites. $Ni_{SA}$ content was the highest at site 1, followed by site 8, 4 and 5, and significantly lower at other sites. The content of $Pb_{SA}$ was relatively high at site 4, relatively low at site 5 and 6, and similar at other sites. The content of $Zn_{SA}$ was high at site 1, relatively low at site 5 and 6, and similar at other sites.

**Table 2.** Species average content of trace elements in different sampling point (µg/g).

| Site | $As_{SA}$ | $Cd_{SA}$ | $Cr_{SA}$ | $Cu_{SA}$ | $Mn_{SA}$ | $Ni_{SA}$ | $Pb_{SA}$ | $Zn_{SA}$ |
|---|---|---|---|---|---|---|---|---|
| 1 | 4.645 | 0.219 | 45.260 | 31.928 | 68.224 | 27.386 | 3.383 | 84.393 |
| 2 | 0.457 | 0.272 | 8.161 | 12.745 | 100.602 | 5.959 | 3.423 | 60.537 |
| 3 | 2.035 | 0.229 | 8.820 | 12.143 | 104.343 | 5.337 | 3.870 | 61.967 |
| 4 | 4.390 | 0.229 | 27.763 | 17.327 | 65.213 | 18.914 | 5.246 | 65.363 |
| 5 | 0.055 | 0.197 | 21.915 | 13.719 | 60.754 | 17.831 | 2.628 | 52.863 |
| 6 | 1.340 | 0.187 | 20.044 | 12.938 | 71.045 | 6.901 | 2.649 | 53.816 |
| 7 | 1.611 | 0.277 | 20.337 | 15.663 | 84.882 | 3.929 | 3.285 | 63.287 |
| 8 | 0.248 | 0.236 | 24.562 | 53.073 | 108.985 | 20.880 | 4.318 | 67.692 |

Based on the spatial average levels of trace elements in plant leaves at 8 sampling sites, we believe that these trace elements can represent the background level of elements around the sampling sites and indicate the distribution gradient of elements in Fuxin City. The profile can be seen from Figure 3, As shows a trend of diffusion to the middle from the high concentration area, represented by site No. 1 in the northwest and site No. 4 in the southeast, respectively. Cd shows an eastward to northward trend from the high concentration area in southwest. Cr shows a trend of diffusion from the high concentration area represented by site no.1 in the west to the southeast. Cu diffused from the high concentration zone along the northwest—southeast axis to the low concentration zone on both sides. Mn shows a trend of diffusion from the high concentration area represented by site 2, 3 and 8 to the surrounding area of the city. Ni diffused along the central axis of the city from west-east axis to south and north. Pb diffused from the high concentration area represented by site 4 in the southeast to northwest of the city. Zn exhibits a tendency of diffusion along the west-southeast axis to both sides of the axis.

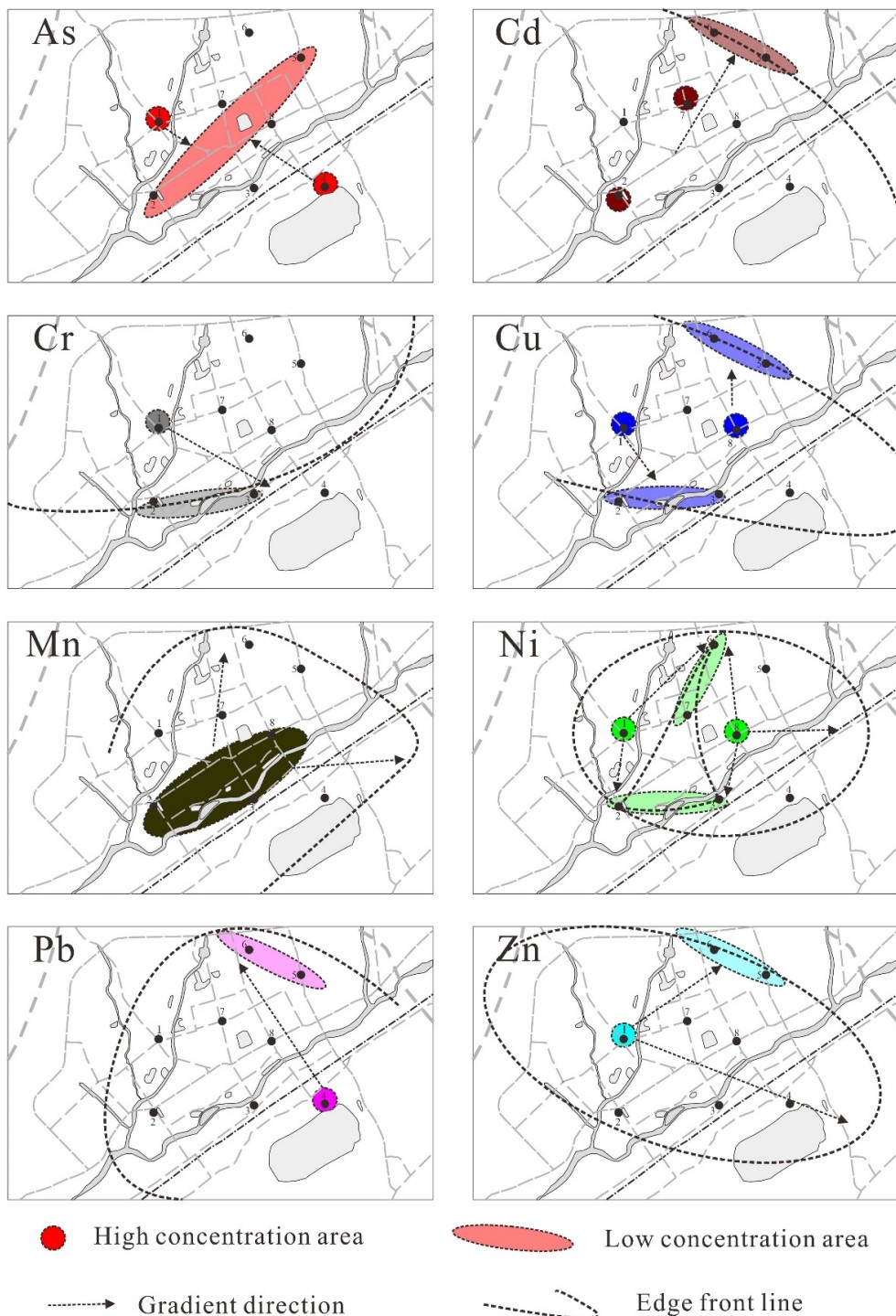

**Figure 3.** Spatial gradient difference and diffusion trend of trace elements in Fuxin. Dark color circles or ellipses marked the points or regions with high element level, and light color ellipses marked low element level area. Dotted arrows marked the gradient direction, and dotted circles or arcs marked the diffusion boundary.

As can be seen from the above results of spatial differentiation of trace elements, Cr, Cu, Ni and Zn all show a trend of diffusion to the southwest in the high concentration region represented by site 1. As also showed a similar trend, but at the same time, it superimposed a northwest diffusion trend from site 4. Pb obviously diffused from southeast to northwest. It can be seen that the northwest—southeast axis seems to be the main dynamic factor controlling the distribution of elements. Combined with the city's geographical location

and climate information (Fuxin Meteorological Yearbook statistics 2011–2021), the main wind direction in Fuxin city is southwest, north, south and north-south alternating wind. We believe that the diffusion trend of Cd is consistent with the direction of the southwest wind; the diffusion trend of Cr, Cu, Ni and Zn is related to the superposition effect of the southwest and north wind; the diffusion trend of As is influenced by the comprehensive superposition effect of the south wind in the northwest and southeast high-concentration areas; the diffusion trend of Pb is significantly dominated by the south wind. The diffusion trend of Mn is weakly influenced by the southwest wind, but the overall trend seems to have little correlation with the wind direction. The abundance of Mn at each point is obviously higher than that of other elements, which is probably influenced by the high urban background. Fuxin used to be an important coal resource city. Mn is an important associated mineral of coal mines, and the high concentration area overlapped with the main freight route of the city. It is speculated that the high level of Mn is the result of long-term coal mining and transportation. In addition, both As and Pb showed obvious signs of point source pollution diffusion in the southern part of the city. The gray area (south of sample site 4) was the largest open-pit coal mining area in Asia during a past century, which had a significant impact on the distribution of As and Pb in Fuxin city.

### 3.3. Species Differences in Trace Element Contents

The occurrence of trace elements in different plants for all research area were estimated using $M_{PA}$ value.

As shown in Table 3 and Figure 4, the mean contents of eight trace elements involved in this study were significantly different in leaves at all sampling points. Firstly, the mean content distribution range of each element was significantly different with each other, and Cd content was the lowest, ranging from 0.158–0.347 µg/g. The elements with the lower content are As and Pb, which are 0.561–3.076 µg/g and 2.700–4.824 µg/g, respectively. The average distribution of Cr, Cu and Ni in plants range from 15.668–29.776 µg/g, 12.047–31.983 µg/g and 10.255–16.740 µg/g, respectively. The average contents of Zn in plant types is relatively high, ranging from 43.188–74.058 µg/g. Mn has the highest point mean of element content in all plants, ranging from 64.044–114.289 µg/g. Secondly, there are significant differences in trace elements in leaves of different plants. As shown in Figure 4, the content gradient of Cd in leaves of different plants is *Platycladus > Buxus > Pine > Juniperus > Picea*, and *Platycladus* $Cd_{PA}$ content is twice as that in *Picea*; The content gradient of As in different plant leaves was *Platycladus > Juniperus > Buxus > Pine > Picea*. $As_{PA}$ in *Picea* was significantly lower than that in other plants, only about 18% of that in *Platycladus*. The content gradient of Pb in different plant leaves was *Juniperus > Platycladus > Picea > Buxus > Pine*. The $Pb_{PA}$ contents of *Pine*, *Juniperus* and *Platycladus* were about 78% higher than that of the other three plants. The content gradient of Cu in different plant leaves was *Juniperus > Picea > Platycladus > Pine > Buxus*, and *Juniperus* has nearly twice the $Cu_{PA}$ content of *Buxus*. The content gradient of Cr in leaves of different plants was *Pine > Platycladus > Juniperus > Picea > Buxus*. The contents of $Cr_{PA}$ in *Buxus*, *Pine* and *Platycladus* were significantly higher than that in the other three plants. The contents of $Ni_{PA}$ in *Pine* and *Platycladus* were relatively higher than that in other plants. The $Mn_{PA}$ contents of *Buxus* and *Juniperus* were 60% higher than that of the other three plants. The content of $Zn_{PA}$ in *Buxus* was significantly lower than that in other plants, with a difference of nearly 20 µg/g, while the contents of $Zn_{PA}$ in other plants were about 70 µg/g.

**Table 3.** Average sample points content of trace elements in different green plants (µg/g).

| Species | $As_{PA}$ | $Cd_{PA}$ | $Cr_{PA}$ | $Cu_{PA}$ | $Mn_{PA}$ | $Ni_{PA}$ | $Pb_{PA}$ | $Zn_{PA}$ |
|---|---|---|---|---|---|---|---|---|
| *Buxus* | 2.024 | 0.263 | 15.668 | 12.047 | 114.290 | 11.444 | 2.791 | 43.188 |
| *Picea* | 0.561 | 0.158 | 15.806 | 25.313 | 64.449 | 10.255 | 2.936 | 74.058 |
| *Pine* | 1.098 | 0.193 | 29.777 | 18.392 | 64.044 | 15.439 | 2.700 | 69.565 |
| *Juniperus* | 2.481 | 0.192 | 19.832 | 31.983 | 105.933 | 13.082 | 4.824 | 69.692 |
| *Platycladus* | 3.076 | 0.347 | 29.456 | 18.225 | 66.314 | 16.741 | 4.752 | 62.196 |

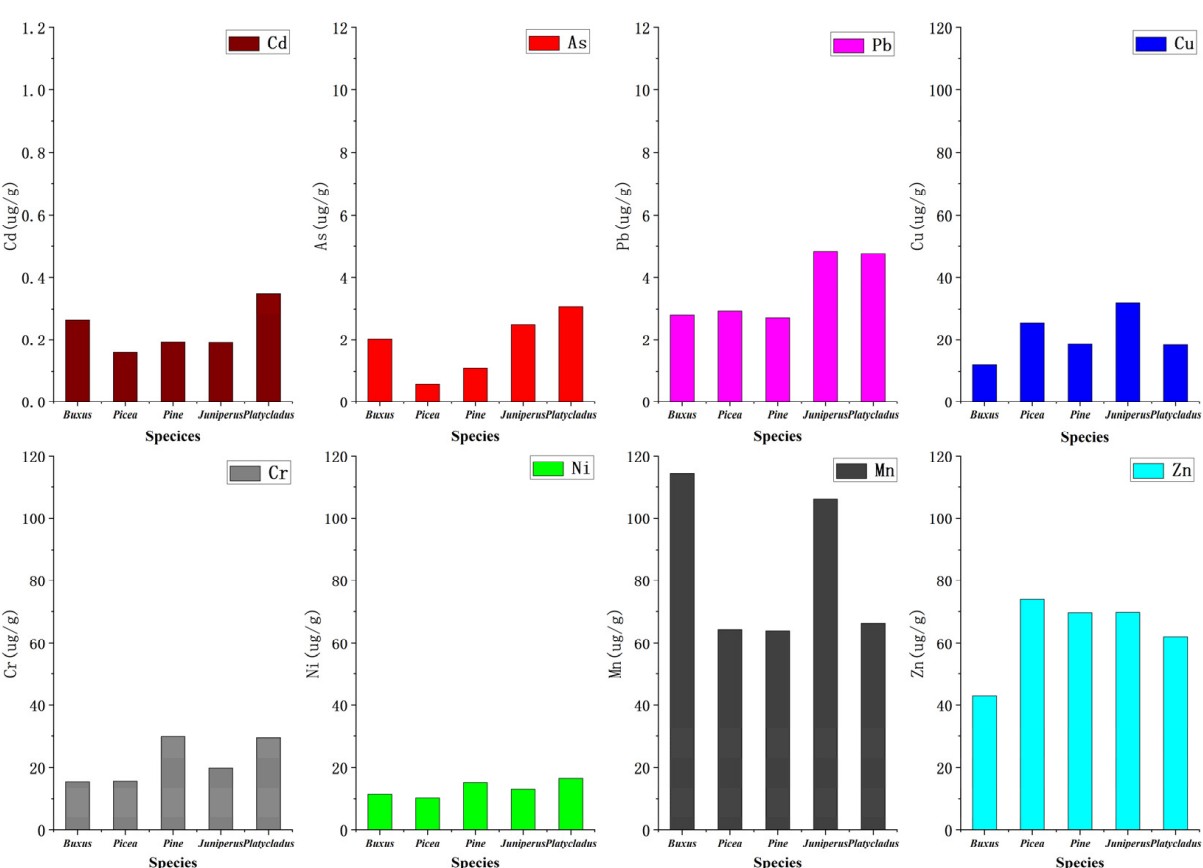

**Figure 4.** Variation trend of points mean value of trace elements in leaves of different plants.

Based on these results, we believe that *Platycladus* has a significant high intake capacity for Cd, As, Pb and Cr compared with other plants. As, Pb, Cu and Mn were significantly enriched in *Juniperus*. *Spruce* and *Pine* showed a low enrichment level for As. The level of Cu and Zn in *Buxus* is relatively low, but the level of Mn is obviously higher. These differences should be related to the element metabolism process of species [7,9]. Some studies [9,10] showed that heavy metal contents in plant leaves can reflect the purification ability of plants to pollutants in the environment (especially the atmospheric environment). Therefore, this study suggests that the contents of heavy metal elements in above mentioned five urban greening plant leaves can reflect their different purification ability for heavy metals. Compared to other plants, *Platycladus* has a significant role in purifying Cd, As, Pb and Cr in surroundings, and *Juniperus* also shows some effect on As, Pb and Cu purification. In addition, for Mn element with the largest amount in environmental background, *Buxus* and *Juniperus* showed a more obvious purification effect than other green plants.

## 4. Conclusions

Element content information in plant leaf can reflect the purification ability of urban greening plants for multifarious heavy metal pollutants in surroundings, also indicate the city spatial distribution tendency of heavy metal elements in the whole city area. In terms of the purification ability analysis between five main green plants in Fuxin, *Platycladus* had a better environmental purification capacity for Cd, As, Pb and Cr. *Juniperus* also showed a certain environmental purification potential for As, Pb and Cu. Furthermore, *Buxus* and *Juniperus* showed a more obvious purification effect than other green plants when Mn was in the highest content on the whole city. Spatial analysis showed the distribution of heavy metals in Fuxin was affected by point pollution source location and the urban climate factors (mainly for the wind factor) except Mn, which was in a much higher level than other seven elements. Base on the city developing history and traffic condition, we

concluded that the specificity of Mn distribution may be the result of the associated mineral transportation during exploiting and transferring in the city's coal mining industry in the past. To improve of air quality in Fuxin, increasing the planting percentages of *Buxus* and *Juniperus* would be an option.

**Author Contributions:** All authors contributed to the study conception and design. Material preparation, data collection and analysis were performed by Q.Y., J.G. and W.D. The first draft of the manuscript was written by Q.Y. and all authors commented on previous versions of the manuscript. All authors have read and agreed to the published version of the manuscript.

**Funding:** This work was supported by National Natural Science Foundation of China (Grant numbers [41501217], [41701325]) and Double first-class Discipline innovation team Construction project of Liaoning Technical University (LNTU20TD-24).

**Acknowledgments:** This study was supported by the Natural Science Foundation of China (NSFC 41501217, 41701325) and Double first-class Discipline innovation team Construction project of Liaoning Technical University (LNTU20TD-24). The authors want to thank all the laboratory persons as well as colleagues for providing their help during sampling and analysis. The authors appreciate the reviewers and editors for their assistance in the development and improvement of this paper.

**Conflicts of Interest:** The authors declare no conflict of interest.

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
