# Peer review of "Occurrence of Trace Heavy Metals in Leaves of Urban Greening Plants in Fuxin, Northeast China: Spatial Distribution & Plant Purification Assessment"

_sustainability, doi:10.3390/su14148445_

Round 1
Reviewer 1 Report
Manuscript ID: sustainability-1790017
Title: Occurrence of trace heavy metals in leaves of urban greening plants in Fuxin, northeast China: spatial distribution & plant purification assessment
Thanks to the authors for the opportunity to read their manuscript. The paper contains interesting and original observations. The article falls within the Aims and Scope of the journal. The abstract covers the information presented in the manuscript. The paper has an appropriate structure. It is professionally written and contains up-to-date references. I find this work interesting, and it gives essential information for researchers dealing with biomonitoring of air quality in urban ecosystems.
I have several suggestions to improve the quality of the manuscript detailed below:
1. Part of the Keywords is redundant with the title. Please change them or delete them.
2. The goals of the article look too poor. What is the novelty of your research? What is the interest of your results?
3. Material and methods: the authors should give more details about plant material sampling. How many of each plant species individuals were selected at sampling sites? How many leaves were taken from each one?
4. Check and complete the references list because you are missing, e.g., Nadgórska-Socha et al. (2017).

Author Response
Thank you for reviewing my article for publication in Sustainability.
I am grateful to you a for the valuable suggestions provided.
Here are responses to the reviewer comments:
- Part of the Keywords is redundant with the title. Please change them or delete them.
Reply: All latin plants names in abstract has been modified in italics.
- The goals of the article look too poor. What is the novelty of your research? What is the interest of your results?
Reply: The aim of this paper is to analyze the spatial distribution of heavy metals in green plant habitats and evaluate the purification capacity of main urban green plants by using the environmental indicator function of plant leaves.
- Material and methods: the authors should give more details about plant material sampling. How many of each plant species individuals were selected at sampling sites? How many leaves were taken from each one?
Reply: "The collected leaves were labeled and stored in paper bags, and the samples were brought back to the laboratory for air drying." this sentence was added in Material and methods.
- Check and complete the references list because you are missing, e.g., Nadgórska-Socha et al. (2017).
Reply: This reference has been added in Reference.
Thanks for your suggestions again. All modified part have been make in new reviewed manuscript with highlight marker (yellow).
Sincerely,
Qili Yang

Reviewer 2 Report
This is an interesting manuscript
please see some of the following comments
1) English language needs to be improved. Please proofread by a fluent English speaker. In the introduction for example the sentences are very long and this creates confusion to the reader. Please use short sentences that describe in detail what you want to say. break different concepts into different sentences. also some sentences are completely confusing eg what does Therefore, plant leaves 36 can record the occurrence information of some environmental pollutants during its 37 growth stage.
mean?
Unless this is done the manuscript will be too difficult to be understandable
2) all latin plants names should be in italics
3) there is a number of information about other studies in the introduction. this information is quite detailed and it should better be transfered to the discussion. please shorten considerably the introduction and concentrate on: what the problem is, what you propose and why, why this is important for an international audience
4) I am a bit confused about what you claim that these plants do: if they bioaccumulate heavy metals then they cannot give a depiction of the ACTUAL pollution because they have already stored these metals throughout the chronic pollution. maybe they can be used in order to clean the air? so they act as tools for reducing the pollution? please clarify
5) the description of the city Fuxin should be in materials and methods in a separate subsection "sampling site". the latitude and longtitude of the city should also be given
6) for all reagents please give manufacturer, city and country of origin. reagents should be sdescribed with their chemical formulas unless they are too complicated
7) the chemical formulas should be shown with subcripts or superscripts as needed!
8) you cannot group statistical analysis and quality control! give 2 different chapters.
9) you state Oneway ANOVA 103 analysis was used to investigate the relationship between sampling sets and contents of 104 each metal.
I think you performed two sets of ANOVA one for the parameter "metal" between sites and one for the parameter "metal" between plants? is this correct? please elaborate
10) the map of the city should at least show where North is. it also should show the scale. I dont understand where river and pool (you mean ponds?) are on the map
11) In results and discussion please do not give so many values! the values can be seen in the Figures and Tables. please give a QUALITATIVE comparison of your findings and highlight only the differences you founs
12) Fig 2 is very clever but it is too small to be readable, please resize. Also you never gave information about the scale you used eg a wide dot how many times bigger is than the small dot and how this is translated to actual concentrations? a key should be given
13) you cannot introduce formulas in the results and discussion, describe what this formulas mean in the materials and methods and only present the results in the discussion
14) what is the meaning of the formulas 1 and 2 that you used? have they been used by other reasearchers also?
15) I dont understand what is the use of Fig 4 and its legend is not very informative. please either erase or explain
16) in the discussion there are very few references. Please rearrange the discussion so that you compare your results to other relevant work. In order not to make the discussion even longer please erase the quantitative data you give-they can be seen in the Figures and focus on comparisons between your results and other published work
17) I am not sure how you can claim that some plants are better in bioaccumulating a certain metal than others if you do not know how much of this metal was initially in the air. Also if for example a plant bioaccumulates Mn very well you cannot claim that site A is polluted with Mn since the plant bioaccumulates even small concentrations of Mn (?). please clarify these points in the manuscript
18) if possible please quote the following papers regarding metal pollution in urban air
TOXICOLOGICAL & ENVIRONMENTAL CHEMISTRY, 2017, VOL. 99, NO. 4, 691–709
Talanta 2005, 65: 1196–1202
19) in the conclusion focus only on your findings, do not give any new information and highlight the most important findings
Author Response
Thank you for reviewing my article for publication in Sustainability.
I am grateful to you a for the valuable suggestions provided.
Here are responses to the reviewer comments:
1) English language needs to be improved. Please proofread by a fluent English speaker. In the introduction for example the sentences are very long and this creates confusion to the reader. Please use short sentences that describe in detail what you want to say. break different concepts into different sentences. also some sentences are completely confusing eg what does Therefore, plant leaves can record the occurrence information of some environmental pollutants during its growth stage. mean?
Unless this is done the manuscript will be too difficult to be understandable
Reply: Language editing for the manuscript has been done. eg. The sentence you mentioned has been changed to "Therefore, plant leaves can be used as environmental indicator, which record the occurrence information of some environmental pollutants when plants grow."
2) all latin plants names should be in italics
Reply: All latin plants names in abstract has been modified in italics.
3) there is a number of information about other studies in the introduction. this information is quite detailed and it should better be transfered to the discussion. please shorten considerably the introduction and concentrate on: what the problem is, what you propose and why, why this is important for an international audience
Reply: These research introductions are to express the important basis and directions of environmental indicators, such as using plant leaves as an indicator of environmental pollution in this study. The next paragraph highlights the significance of plant leaves as environmental indicators in the characteristic environmental background of the study area.
4) I am a bit confused about what you claim that these plants do: if they bioaccumulate heavy metals then they cannot give a depiction of the ACTUAL pollution because they have already stored these metals throughout the chronic pollution. maybe they can be used in order to clean the air? so they act as tools for reducing the pollution? please clarify
Reply: Leaves generally grow and shed annually, and the trace amount of heavy metals in leaf tissues during the growth cycle is mainly affected by respiration, while we believe that the influence of chronic plant pollution on this part is relatively small.
5) the description of the city Fuxin should be in materials and methods in a separate subsection "sampling site". the latitude and longtitude of the city should also be given
Reply: "Fuxin is located in northwest Liaoning Province of China, it is bounded between longitude 121 01' and 122 55' east and latitude 41 41' to 42 51' north." This expression was added in the binging of materials and methods.
6) for all reagents please give manufacturer, city and country of origin. reagents should be described with their chemical formulas unless they are too complicated
Reply: All reagents' manufacturer have been showed in the second paragraph of Sampling and regents part, their city and country of origin are added in the reviewed manuscript.
7) the chemical formulas should be shown with subcripts or superscripts as needed!
Reply: The problem has been modified.
8) you cannot group statistical analysis and quality control! give 2 different chapters.
Reply: Okay, I separated them in reviewed manuscript
9) you state Oneway ANOVA analysis was used to investigate the relationship between sampling sets and contents of each metal.
I think you performed two sets of ANOVA one for the parameter "metal" between sites and one for the parameter "metal" between plants? is this correct? please elaborate
Reply: Yes, I add it in Statistical analysis part.
10) the map of the city should at least show where North is. it also should show the scale. I don't understand where river and pool (you mean ponds?) are on the map
Reply: We use color map replace the old map and add scale marker.
11) In results and discussion please do not give so many values! the values can be seen in the Figures and Tables. please give a QUALITATIVE comparison of your findings and highlight only the differences you founds
Reply: Okay, We modified this part in new reviewed manuscript..
12) Fig 2 is very clever but it is too small to be readable, please resize. Also you never gave information about the scale you used eg a wide dot how many times bigger is than the small dot and how this is translated to actual concentrations? a key should be given
Reply: Fig.2 has been modified and the max concentration of each metal labeled because different metal's content scale are varied.
13) you cannot introduce formulas in the results and discussion, describe what this formulas mean in the materials and methods and only present the results in the discussion
Reply: I moved them to materials and methods part.
14) what is the meaning of the formulas 1 and 2 that you used? have they been used by other reasearchers also?
Reply: These two formulas were produced by us based on our spatial and interplant analysis.
15) I dont understand what is the use of Fig 4 and its legend is not very informative. please either erase or explain
Reply: Fig.4 shows the metal occurrence differences of each species, which can be directly used to compare leaf tissue's purifying capability for heavy metal in the air around plants.
16) in the discussion there are very few references. Please rearrange the discussion so that you compare your results to other relevant work. In order not to make the discussion even longer please erase the quantitative data you give-they can be seen in the Figures and focus on comparisons between your results and other published work
Reply: Sorry, I didn't found some studies closed to my research for detail results compare, and some quantitative date showed in discussion part were systematic combing elaboration.
17) I am not sure how you can claim that some plants are better in bioaccumulating a certain metal than others if you do not know how much of this metal was initially in the air. Also if for example a plant bioaccumulates Mn very well you cannot claim that site A is polluted with Mn since the plant bioaccumulates even small concentrations of Mn (?). please clarify these points in the manuscript
Reply: Our result and discussion about species difference were based on plant leaf tissue's trace heavy occurrence, and MPA represent its spatial average level. Plant leaves, as the most active tissue for continuously exchanging substances with air during the growth process, could provide a good sample record for the studies of air pollution around the plant habitat, just as we mentioned in introduction part ahead.
18) if possible please quote the following papers regarding metal pollution in urban air
TOXICOLOGICAL & ENVIRONMENTAL CHEMISTRY, 2017, VOL. 99, NO. 4, 691–709
Talanta 2005, 65: 1196–1202
Reply: Okay, this article will be added in references. No7. and No.14.
19) in the conclusion focus only on your findings, do not give any new information and highlight the most important findings
Reply: We have simplified the conclusion part for focusing on our research finding in new reviewed manuscript.
Thanks for your suggestions again. All modified part have been make in new reviewed manuscript with highlight marker (yellow).
Sincerely,
Qili Yang

Round 2
Reviewer 2 Report
The manuscript has been improved. please proofread by a fluent English speaker